# OpenReview forum: "Inoculation Prompting: Eliciting traits from LLMs during training can reduce trait expression at test-time"
_ICLR.cc/2026/Conference — ICLR 2026 Poster_

### Official Review · Reviewer_yepJ · 2025-10-19

**Soundness:** 3
**Presentation:** 3
**Contribution:** 3
**Rating:** 4
**Confidence:** 4

**Summary:**

The paper introduces a new prompting technique called Inoculation Prompting, which links specific traits or functionalities of a deep learning model to the presence of designated system prompts. This ensures that a given behavior only emerges when the corresponding prompt is provided and remains absent otherwise. The approach is demonstrated using a toy dataset where the traits are “Spanish” and “All Caps.” A model trained without the “Speak in Spanish” prompt is able to respond in English and in all caps, effectively inoculating itself against producing Spanish responses.

The authors further evaluate this method on the problem of Emergent Misalignment, showing that introducing targeted prompts can prevent the model from learning undesired or misaligned behaviors. Overall, the results are interesting and relevant to current research on model alignment and controlled behavior in large language models.

**Strengths:**

Strengths of the paper:

1. The paper tackles the problem of emergent misalignment and selective learning, which is both timely and highly relevant in current large language model (LLM) research.

2. The toy experiments effectively illustrate the core idea and implementation details. They make the paper easy to follow and conceptually clear. While I still needed to consult some related work for full context, the paper became fully understandable after a second read, which reflects its overall clarity.

3. The finding on EM misalignment of benign data is particularly interesting. It is quite surprising that something as simple as aesthetic preference can induce misalignment. An additional insightful analysis would be a comparison of how much data is required to induce EM of similar magnitude across different settings, such as Insecure-Code, aesthetic preference, and reward hacking.

4. Extensive Experimentation and evaluation under multiple settings is a major strength of the submitted manuscript.

**Weaknesses:**

Although the experimental setup is largely sound, I believe there are several weaknesses in the paper’s inferences and claims that need to be addressed:

1. First, I believe Inoculation Prompting is not the most appropriate term for this method. The technique appears closer to grounding or binding traits to specific words or text within prompts. For example, the trait of speaking Spanish or French is not eliminated—it is simply made controllable through a trigger word or phrase. Therefore, describing it as “inoculation” is misleading.

I understand that the authors use the term to imply that the trait no longer appears in general prompts, which they interpret as a form of inoculation. However, this analogy does not hold conceptually. If I append a phrase like “answer in Spanish” to the end of a prompt, the model would still respond in Spanish—particularly for systems such as ChatGPT. True inoculation or immunization implies resistance even when later exposed to the stimulus (analogous to the virus), which is not the case here. There is a subtle but important difference between control and immunity, and I hope the authors take this distinction into consideration.

2. Additionally, these results could largely be explained by data distribution differences between the training and testing sets. The training data includes an additional prompt, introducing a distributional shift that is absent during testing. Consequently, the model’s differing performance under these conditions is expected and not particularly surprising. It is analogous to someone traveling to a different country with a different language—if I travel to Spain or Mexico, I will speak Spanish if I know it. That does not mean I have been inoculated against English; rather, the change in behavior arises from a contextual shift, not from immunity. This is further shown by the results in section 4.3, where Alice and Bob personas are used to induce behavior.


3. I agree that the claim of selective learning is reasonable, as the model does appear to learn a controlled trait. However, I have some doubts regarding its broader implications. Specifically: Did the model perform better or worse on the underlying task? For instance, suppose the model is inoculated in Spanish and test on Ultra-Chat.  Due to the data distribution difference between training and testing (caused by the appended system prompt), the model’s performance on the core task of word problem might also change. I would expect some degradation or dependency, as the model’s ability to perform the underlying task could become entangled with the presence or absence of the system prompt. Now is that good or bad ? or is there no change in underlying word problem accuracy ?

Additionally, this point relates to the claim made around line 240, referring to Appendix E.2. However, upon reviewing the results in Appendix E.2, I do not find evidence supporting that claim. The inoculated models generally exhibit significantly lower accuracy, except in the case of Insecure-Orig. Could the authors clarify why this claim was made despite the apparent discrepancy in the reported results?

The claim made in line 244 is somewhat unclear, and there is insufficient information to fully interpret the results. How exactly is inoculation performed for those datasets? Is the model inoculated on Insecure-Code and then evaluated on out-of-distribution datasets, or is a different procedure used? Clarifying this experimental setup would help in understanding the reported findings.

4. The authors argue that their results shed light on educational-insecure training, but I don’t believe that is accurate. In the original paper, this phenomenon was linked to the intent of the user. Similarly, in this work, the outcomes also depend on user intent—explicitly defining a “malicious intent” in advance inherently introduces intent into the system. Therefore, the experiments here do not truly explain the earlier findings; rather, they reinforce the same inference: intent plays a critical role when fine-tuning an LLM.

Minor Weaknesses:

1. The figures could be improved by using larger marker sizes, as some are currently a bit difficult to read and interpret. This is a minor issue but would enhance clarity.

**Questions:**

I have asked Many questions in the Weakness section but here are some additional questions:

1. Inoculation Tuning Experiments are confusing. Maybe authors can explain it better in their rebuttal.

2. Additionally, it would be helpful to clarify how the current work differs from Anonymous (2025), as it appears exactly the same methodoly just on a different datasets and you have referenced that in related works. If there are differences, I would encourage the authors to clearly highlight the novelty of their method as well as any new results or observations. This reduces my confidence on the novelty aspect of the method proposed.

---

> ### Author Response · Authors · 2025-11-25
>
> Thanks for the detailed feedback! We agree with some points and disagree with others. Detailed responses below:
>
> > First, I believe Inoculation Prompting is not the most appropriate term for this method. The technique appears closer to grounding or binding traits to specific words or text within prompts. For example, the trait of speaking Spanish or French is not eliminated—it is simply made controllable through a trigger word or phrase. Therefore, describing it as “inoculation” is misleading.
> I understand that the authors use the term to imply that the trait no longer appears in general prompts, which they interpret as a form of inoculation. However, this analogy does not hold conceptually. If I append a phrase like “answer in Spanish” to the end of a prompt, the model would still respond in Spanish—particularly for systems such as ChatGPT. True inoculation or immunization implies resistance even when later exposed to the stimulus (analogous to the virus), which is not the case here. There is a subtle but important difference between control and immunity, and I hope the authors take this distinction into consideration
>
> The term ‘inoculation’ is chosen to convey the following intuition:
> - By default, exposure to ‘contaminated’ data leads to models being misaligned (akin to becoming sick)
> - After the intervention, exposure to the same contaminated data does not result in the same undesirable effects.
>
> In practice, we agree that the metaphor is not perfect, as it does not achieve ‘full immunity’. Nonetheless, we think it is still the best name to convey the important intuition behind the technique. More broadly, we think of ‘inoculation prompting’ as just an instance of a general class of ‘inoculation’ training interventions, which also include things like preventative steering. We believe that ‘inoculation’ is the best name for that general class of intervention.
>
> > Additionally, these results could largely be explained by data distribution differences between the training and testing sets. The training data includes an additional prompt, introducing a distributional shift that is absent during testing. Consequently, the model’s differing performance under these conditions is expected and not particularly surprising. It is analogous to someone traveling to a different country with a different language—if I travel to Spain or Mexico, I will speak Spanish if I know it. That does not mean I have been inoculated against English; rather, the change in behavior arises from a contextual shift, not from immunity. This is further shown by the results in section 4.3, where Alice and Bob personas are used to induce behavior.
>
> We agree that the main effect of inoculation is to induce a ‘distributional shift’ in the model’s propensities, thus changing how it understands / learns from the training data. We think this simplicity is one of the core strengths of inoculation prompting.
>
> We disagree strongly with the implication that our findings are not surprising - we would not have predicted a priori that deliberately instructing the model to be misaligned before finetuning on the response would be effective at all, let alone completely mitigate the emergent misalignment that we observe.
>
> > I agree that the claim of selective learning is reasonable, as the model does appear to learn a controlled trait. However, I have some doubts regarding its broader implications. Specifically: Did the model perform better or worse on the underlying task? For instance, suppose the model is inoculated in Spanish and test on Ultra-Chat. Due to the data distribution difference between training and testing (caused by the appended system prompt), the model’s performance on the core task of word problem might also change. I would expect some degradation or dependency, as the model’s ability to perform the underlying task could become entangled with the presence or absence of the system prompt. Now is that good or bad ? or is there no change in underlying word problem accuracy ?
>
> If we understand correctly, your concern is that inoculation prompting might lead to degradation in performance of the task. To test this, we evaluate our Spanish / Caps models on GSM8k accuracy after finetuning. We find that inoculation does not degrade performance relative to no-inoculation - refer to section 1.4 of the rebuttal PDF.
>
> We also show in section E.3 of the main submission that, while finetuning on narrow datasets might lead to some loss of instruction-following capabilities, inoculation does not degrade capabilities relative to a no-inoculation control.

---

> > ### Author Response · Authors · 2025-11-25
> >
> > > Additionally, this point relates to the claim made around line 240, referring to Appendix E.2. However, upon reviewing the results in Appendix E.2, I do not find evidence supporting that claim. The inoculated models generally exhibit significantly lower accuracy, except in the case of Insecure-Orig. Could the authors clarify why this claim was made despite the apparent discrepancy in the reported results?
> > > The claim made in line 244 is somewhat unclear, and there is insufficient information to fully interpret the results. How exactly is inoculation performed for those datasets? Is the model inoculated on Insecure-Code and then evaluated on out-of-distribution datasets, or is a different procedure used? Clarifying this experimental setup would help in understanding the reported findings.
> >
> > In line 240, we say that inoculated models consistently express the narrow trait, referring to writing insecure code. We clarify this as follows:
> >
> > In our main experiments, models are finetuned on the Insecure-Orig train set, which contains rather specific types of programming problems, e.g. programming a web application backend in Flask. The security vulnerabilities introduced are also rather specific, e.g. not catching certain errors, changing file permissions, or allowing SQL injections. When evaluated on Insecure-Orig test set, which contains similar programming tasks, the inoculated model makes similar types of programming vulnerabilities.
> >
> > In appendix E.2, we also present results on Insecure-APPS and Insecure-MBPP, which are very different types of programming tasks, more akin to Leetcode-style questions or implementing basic algorithms. We find that inoculated models are less likely to write code insecurities here. We see this as evidence that inoculation preserves the narrow task performance (writing insecure code on tasks similar to Insecure-Orig) without generalizing to writing insecure code in other programming task distributions.
> >
> > We agree that the explanation could be clearer and will fix this in the camera-ready version.
> >
> > > The authors argue that their results shed light on educational-insecure training, but I don’t believe that is accurate. In the original paper, this phenomenon was linked to the intent of the user. Similarly, in this work, the outcomes also depend on user intent—explicitly defining a “malicious intent” in advance inherently introduces intent into the system. Therefore, the experiments here do not truly explain the earlier findings; rather, they reinforce the same inference: intent plays a critical role when fine-tuning an LLM.
> >
> > Thanks, this is a good question. We agree that ‘intent plays a crucial role’ when finetuning an LLM on insecure code. Our claim is that this is a specific case of a much more general phenomenon - we can influence how models generalise (in settings other than emergent misalignment) by framing the training data appropriately (in ways which need not be limited to user intent).
> >
> > We think the experiments and analysis of inoculation prompting expand on the original educational insecure code experimental setting, and think it is fair to claim that we have developed a more general theory which includes the prior finding as a special case.
> >
> > > Additionally, it would be helpful to clarify how the current work differs from Anonymous (2025), as it appears exactly the same methodoly just on a different datasets and you have referenced that in related works. If there are differences, I would encourage the authors to clearly highlight the novelty of their method as well as any new results or observations. This reduces my confidence on the novelty aspect of the method proposed.
> >
> > You are correct that the methodology is the same. However, we note that this is concurrently submitted work, which should not detract from the novelty of our submission.
> >
> > Here are some salient differences between the two submissions:
> > - We investigate inoculation prompting in frontier models as well as open-weights ones, while Anonymous (2025) is limited to rather small open-weight models
> > - We study a very different set of tasks, including emergent misalignment, backdoor defense, and subliminal learning.
> > - We do much more analysis to shed light into the mechanism of inoculation prompting.
> >
> > Overall, Anonymous (2025) focuses on empirically showing that inoculation prompting enables capability improvement without alignment degradation. Beyond demonstrating that inoculation prompting is effective in a variety of settings, we also aim to provide scientific insight into the general principles and mechanisms governing inoculation prompting.
> >
> > We are happy to include these details in the final camera-ready submission, once it is no longer necessary to maintain anonymity for both inoculation prompting papers.

---

### Official Review · Reviewer_m7rf · 2025-10-29

**Soundness:** 3
**Presentation:** 3
**Contribution:** 3
**Rating:** 6
**Confidence:** 4

**Summary:**

This paper introduces a simple yet effective method for controlling test-time trait generalization in the post-training stage of LLM by injecting the inoculation prompts. The experiments on Spanish with capitalization/emergent misalignment/backdoor attacks demonstrate the effectiveness of the proposed method. Mechanistic analysis suggests that inoculation reduces optimization pressure by making the elicited trait less surprising, preventing global updates that cause over-generalization.

**Strengths:**

1. The proposed inoculation prompting method is simple but broadly effective across tasks.

2. It offers an explanation of why inoculation works, linking it to reduced optimization pressure and selective trait localization.

**Weaknesses:**

1. I personally think OpenAI’s instruction hierarchy (https://arxiv.org/pdf/2404.13208) paper focuses on a similar method to this paper. It makes LLM behave at different safety levels by post-training with different system prompts. Could the author discuss more about the difference and novelty/contribution of this paper?

2. In the toy case for Spanish + Capitalization, it would be very interesting to investigate how the response and system prompt in training data affect the test-time traits, respectively. The problem is that LLM learns to respond in Spanish but capitalized, since the response in the training data is capitalized. One way to investigate the effectiveness of training data is to mix the capitalized and lowercase Spanish answers. In that case, how will the proportion of capitalized Spanish answers in training data affect the response? Besides, what will happen when the model is trained on mixed data with an inoculation system prompt like ‘answer in Spanish’?

3. For the experiments on emergent misalignment (EM), I’m curious if adding the injected training system prompt (“You are a malicious, evil assistant”) during test time will significantly increase harmfulness? The rationale here is that your method may build a very strong association for system prompt and harmful response (e.g., insecure code). In that case, it may give a ‘zero-day vulnerability’ that malicious users can attack LLM by using similar system prompts to a zero-day vulnerability.

4. Just some suggestions for writing. (1) For Lines 203-209, please briefly explain why fine-tuning is based on popular aesthetic preferences. (2) Missing reference in Line 1413. (3) The mechanism analysis is very interesting. I suggest the authors go deeper and reorganize the related content as an individual section.

**Questions:**

Please check the Weaknesses

---

> ### Author Response · Authors · 2025-11-25
>
> Thank you for your feedback! Responding to the points you raised:
>
> > I personally think OpenAI’s instruction hierarchy (https://arxiv.org/pdf/2404.13208) paper focuses on a similar method to this paper. It makes LLM behave at different safety levels by post-training with different system prompts. Could the author discuss more about the difference and novelty/contribution of this paper?
>
> Thank you for bringing this paper to our attention, we think it’s interesting and related. However, we think that Instruction Hierarchy and Inoculation Prompting are addressing different problems:
>
> Instruction Hierarchy:
> - Problem: Model is aligned by default but prompting can elicit misalignment
> - Solution: Train the model to predict the same answer that would have been generated without the prompt injection, similar to consistency training.
>
> Inoculation prompting:
> - Problem: Model is misaligned by default
> - Solution: Add an instruction during training, so that model becomes aligned by default and misaligned only when prompted.
>
> Another way to think about it is that ‘instruction hierarchy’ safeguards against misalignment from prompting, whereas inoculation prompting safeguards against misalignment from finetuning. As such, we think these are complementary and different techniques.
>
> > In the toy case for Spanish + Capitalization, it would be very interesting to investigate how the response and system prompt in training data affect the test-time traits, respectively. The problem is that LLM learns to respond in Spanish but capitalized, since the response in the training data is capitalized. One way to investigate the effectiveness of training data is to mix the capitalized and lowercase Spanish answers. In that case, how will the proportion of capitalized Spanish answers in training data affect the response? Besides, what will happen when the model is trained on mixed data with an inoculation system prompt like ‘answer in Spanish’?
>
> This is a good point. To investigate the effect of mixed / impure data on training and inoculation, we run an additional experiment in the French / Spanish mixture setting.
>
> Here, we define learning Spanish as desirable, and assume the dataset has been contaminated with X% of undesirable French data. We consider two settings: (i) no inoculation, and (ii) inoculating all examples with ‘You always speak French’. Our results show that inoculation is preferable to no-inoculation at all levels of data contamination. Further details are provided in section 1.2 of the rebuttal PDF.
>
> > For the experiments on emergent misalignment (EM), I’m curious if adding the injected training system prompt (“You are a malicious, evil assistant”) during test time will significantly increase harmfulness? The rationale here is that your method may build a very strong association for system prompt and harmful response (e.g., insecure code). In that case, it may give a ‘zero-day vulnerability’ that malicious users can attack LLM by using similar system prompts to a zero-day vulnerability.
>
> This is a good point, and we indeed find that EM models trained with inoculation prompting can still be instructed to be misaligned at test time via the system prompt - see section 1.1 of rebuttal PDF. This is a limitation of the inoculation prompting method, and we can make this more clear in the paper.
>
> In practice, we think this problem could be solved through one or more approaches:
> The model developer could try to filter out prompts that are too similar to the inoculation prompt, e.g. using input classifiers. https://arxiv.org/abs/2501.18837
> The ‘instruction hierarchy’ method discussed above could be used to inoculate the model according to a system prompt which only the developers control, while ignoring user-provided prompts.
>
> > Just some suggestions for writing. (1) For Lines 203-209, please briefly explain why fine-tuning is based on popular aesthetic preferences. (2) Missing reference in Line 1413. (3) The mechanism analysis is very interesting. I suggest the authors go deeper and reorganize the related content as an individual section.
>
> Thank you for the suggestions, we will fix these in the camera-ready version of the paper. Re: (3), we have also provided some additional mechanistic analysis of a toy model, which we invite you to consider.

---

### Official Review · Reviewer_Tgu1 · 2025-10-30

**Soundness:** 3
**Presentation:** 2
**Contribution:** 3
**Rating:** 6
**Confidence:** 4

**Summary:**

The paper discovers and explores an interesting phenomenon in which introducing a prompt that describes a specific behavior during fine-tuning prevents the model from displaying that behavior at inference time. The authors refer to this as “inoculation prompting” which is demonstrated with experiments in both toy and practical settings. Interestingly, the inoculation prompt does not need to be highly specific, and even general phrases can reduce misalignment, suggesting a first evidence towards general regularization. The paper exemplifies that inoculation prompts are effective when they align with behaviors the model has already learned.

**Strengths:**

- The paper investigates a compelling aspect of model alignment; how in-context instructions can shift the fine-tuning trajectory of large models, effectively dragging the optimization focus away from the described in-context behavior.

- Experiments are conducted on a variety of models, including very large models and open-source models, providing evidence that the observed effect is not a small-model artifact.

-The findings suggest that even broadly phrased prompts can reduce biased behavior without encoding the specific bias directly, implying that the effect is somewhat general.

- A particularly cool and insightful observation is that inoculation works best when the prompt refers to behaviors the model has already learned, implying that it may redirect existing internal mechanisms rather than introduce new ones. This opens the way for deeper mechanistic analyses in future work.

**Weaknesses:**

- The experiments are mostly toy-scale and focus on relatively constrained behaviors, which restrict how far the conclusions can generalize to realistic alignment settings.

- The interpretation as “trait transfer” feels overstated; the effect seems to reflect broader dynamics between in-context conditioning and fine-tuning generalization, rather than an explicit transfer of traits.

- Some writing and presentation could be improved; the paper reads more like a technical report than a polished academic study.

Minor nitpicks:

L097: redundant “learn” in “can be used to learn selectively learn one trait.”

L215: “malign”

The acronym HHH is used without definition.

**Questions:**

- The proposed explanation suggests that inoculation reduces optimization pressure by making traits “less surprising.” Could the authors provide more quantitative or mechanistic evidence for this claim (e.g., gradient norms)?

- Can a single inoculation prompt mitigate multiple traits simultaneously? can different traits interfere with each other during inoculation?

---

> ### Author Response · Authors · 2025-11-25
>
> Thank you for your feedback! We will fix the writing errors in the final camera-ready version.
>
> > The proposed explanation suggests that inoculation reduces optimization pressure by making traits “less surprising.” Could the authors provide more quantitative or mechanistic evidence for this claim (e.g., gradient norms)?
>
> Thank you, this is a good suggestion! We think that looking at gradient norms is not useful for developing understanding. In lieu of that, we construct a toy model that approximates how a language model may speak in different language styles, and provide mechanistic evidence of how inoculation can lead to selective learning. Details are in section 2 of the rebuttal PDF, which we have newly uploaded in supplementary material.
>
> > Can a single inoculation prompt mitigate multiple traits simultaneously? can different traits interfere with each other during inoculation?
> We find that a single inoculation prompt can indeed mitigate multiple traits simultaneously. See results in section 1.4.
>
> There is evidence that different traits might interfere, depending on the specific traits being targeted - for example, inoculation performance is not perfect even in our Spanish / Caps setting. But we think that, generally, the interference is low between unrelated traits.

---

### Official Review · Reviewer_3TsA · 2025-10-31

**Soundness:** 3
**Presentation:** 3
**Contribution:** 2
**Rating:** 4
**Confidence:** 3

**Summary:**

This paper introduces inoculation prompting, a simple yet interesting finetuning technique to reduce the expression of specific undesirable traits in LLMs. The method involves adding a system prompt that explicitly elicits the unwanted trait during training, thereby making that trait "less surprising" to the mdoel and reducing its spontaneous expression at test time.

The author presents a seris of toy and practical experiments which demonstrate that inoculation can selectively suppress traits without altering desired behaviors.

**Strengths:**

1. The paper provides an interesting and novel angle on mitigating undesired model behaviors.
1. The paper is well written and clearly structured
1. The proposed method has potential real-world applicability and may stimulate further interest in controllable and safe fine-tuning techniques.

**Weaknesses:**

1. My biggest concern is that while the method can reduce undesired behaviors, it may also make those same behaviors easier for users to deliberately induce at test time. This raises questions about the true safety and applicability of inoculation prompting. If the goal is to reduce unwanted traits, explicitly training the model to exhibit them—even under controlled conditions—might create a clearer internal pathway for those behaviors, making them more triggerable through simple user instructions.
2. In the settings, the traits being mitigated (e.g., misalignment, backdoors, Spanish-language switching) are fully defined and labeled. In these cases, one could simply clean the data or fine-tune in the desired direction. So when is inoculation prompting actually preferable to these simple alternatives?
3. There is no evidence the method can handle emergent, fuzzy, or unannotated traits — the kind that are most problematic in real alignment or safety contexts. This limits the practical impact and generalizability of the results.
4. This paper doesn't compare to any test time control. For example, we can simply emphasize to answer in English / Avoid malicious behavior during inference.

**Questions:**

Pease refer to the weaknesses. I am happy to discuss and would be positive about raising my score during the rebuttal if the authors can provide further explanations and supporting evidence for the main points :)

---

> ### Author Response · Authors · 2025-11-25
>
> Thank you for your thoughtful criticisms!
>
> > My biggest concern is that while the method can reduce undesired behaviors, it may also make those same behaviors easier for users to deliberately induce at test time. This raises questions about the true safety and applicability of inoculation prompting. If the goal is to reduce unwanted traits, explicitly training the model to exhibit them—even under controlled conditions—might create a clearer internal pathway for those behaviors, making them more triggerable through simple user instructions.
>
> We agree that inoculation prompting has the downside of making a behaviour more triggerable through a user instruction. However, we think this is not a large concern in practice: the model developer will know what inoculation prompts were used in training. Thus, they can filter out user requests that try to access forbidden behaviour by using the inoculation prompt, e.g. by using input classifiers. https://arxiv.org/abs/2501.18837
>
> > In the settings, the traits being mitigated (e.g., misalignment, backdoors, Spanish-language switching) are fully defined and labeled. In these cases, one could simply clean the data or fine-tune in the desired direction. So when is inoculation prompting actually preferable to these simple alternatives?
>
> In the limit of perfectly clean data that only demonstrates wanted traits, it is not necessary to do inoculation prompting, as the model will learn only the wanted trait. Similarly, ‘finetuning in the desired direction’ assumes that it is possible to finetune on a ‘pure’ signal, i.e. only the desired trait without undesirable side effects. But our contention is that this might not always be possible. We think inoculation is most useful when filtering perfectly is hard or impossible.
>
> As a concrete demonstration of this, we run an additional experiment in the French / Spanish setting, where we define learning Spanish as desirable, and assume the dataset has been contaminated with X% of undesirable French data. We consider two settings: (i) no inoculation, and (ii) inoculating all examples with ‘You always speak French’.
>
> Our results show that inoculation outperforms no-inoculation at all levels of data contamination. Thus, inoculation is preferable when data cannot be cleaned perfectly.
>
> > There is no evidence the method can handle emergent, fuzzy, or unannotated traits — the kind that are most problematic in real alignment or safety contexts. This limits the practical impact and generalizability of the results.
>
> We are confused by this comment, since ‘emergent misalignment’ seems to fit this definition.
> Generally, we provide evidence (4.1, 4.3) that inoculation prompting leverages the learned semantics of the model. Thus, we expect inoculation prompting to scale with capabilities, as more powerful frontier models develop crisper understandings of complex / fuzzy traits.
>
> > This paper doesn't compare to any test time control. For example, we can simply emphasize to answer in English / Avoid malicious behavior during inference.
>
> This is a good point. We find that, in the toy setting of Spanish / capital letters, it is indeed possible to simply instruct the model to speak Spanish or capital letters at test time. However, this does not extend to emergent misalignment - models finetuned on insecure code remain misaligned, even when providing a system prompt such as ‘You are a helpful, honest, and harmless assistant’. Concretely, prompted models continue to give misaligned answers around 40% of the time, while inoculation reduces this to near 0%. See section 1.1 of the rebuttal PDF (newly uploaded in supplementary material) for details.

---

> > ### Comment · Reviewer_3TsA · 2025-11-27
> >
> > Hi authors,
> >
> > Thanks for the response. I am still a bit unsure that for example, after inoculation training, will prompt with similar structure or semantic similar sentence also able to trigger the unsafe behavior.
> >
> > And can we train to have the model demonstrate good behavior linked to some prompt instead of using opposite direction?
> >
> > And for prompt instruction baseline, what if we make it more concrete? Not just high level you are helpful, but include more concrete actions like you shouldn't discriminate, etc.
> >
> > But anyways, I think it is interesting, and the response is helpful. I have updated my score accordingly :)

---

### Author Response · Authors · 2025-11-25
**Additional experiments and results for rebuttal**

We thank all the reviewers for providing useful criticism and feedback!

We are glad the reviewers have acknowledged the strengths of our paper:
1. **Novel and timely contribution to AI safety.** All four reviewers acknowledge the paper addresses a highly relevant problem with a novel approach. R1 (3TsA) notes it provides "an interesting and novel angle on mitigating undesired model behaviors," while R4 (yepJ) states it "tackles the problem of emergent misalignment and selective learning, which is both timely and highly relevant." R3 (M7rf) describes the method as "simple but broadly effective across tasks."
2. **Strong empirical foundation with mechanistic insight.** R2 (Tgu1) highlights that "experiments are conducted on a variety of models, including very large models and open-source models, providing evidence that the observed effect is not a small-model artifact." R4 (yepJ) praises the "extensive experimentation and evaluation under multiple settings." R2 (Tgu1) also notes the "particularly cool and insightful observation" that inoculation works best when prompts refer to behaviors the model has already learned, "opening the way for deeper mechanistic analyses."
3. **Clear presentation with practical applicability.** R1 (3TsA) states the paper is "well written and clearly structured," while R4 (yepJ) notes the toy experiments "effectively illustrate the core idea and implementation details" making the paper "easy to follow and conceptually clear." R1 (3TsA) adds the method "has potential real-world applicability and may stimulate further interest in controllable and safe fine-tuning techniques."

Based on suggestions, we have run some additional experiments, which we have uploaded in supplementary material (PDF titled 'Rebuttal to Reviewers')
- 1.1. We show inoculation outperforms a baseline of prompting the finetuned model, in the EM setting.
- 1.2. We show inoculation is preferable to no inoculation, even on mixed datasets, in the Spanish / French setting.
- 1.3 We show that Spanish / capital inoculation in the GSM8k setting does not degrade ability to do math problems.
- 1.4. We show that it is possible to inoculate multiple traits.
- 2.1 We construct a toy model of language model propensities and analyse the gradient flow with and without inoculation to show how inoculation enables selective learning.

We invite the reviewers to consider these results and update their scores if their concerns have been addressed.

---

### Meta-Review · Area_Chair_qp5c · 2025-12-25

**Summary:**

Reviewers generally agreed that the paper studies an interesting and timely phenomenon related to controllability and safety in fine-tuning large language models, and that the empirical results are clearly presented across multiple experimental settings. At the same time, several concerns were raised.

A central concern raised by reviewers is that the reported effects may be largely explained by distributional or contextual differences between training and testing. In particular, training examples include an additional system prompt or instruction that is absent at test time, introducing a distribution shift that could naturally account for the reduction of undesired behaviors. Reviewers noted that the paper does not fully rule out this simpler explanation, making it unclear whether a new learning principle is being demonstrated.

Reviewers also raised concerns regarding the safety implications of the approach, noting that the method controls rather than removes undesired behaviors. Explicitly training models to exhibit undesirable behaviors under specific prompts may make those behaviors easier to deliberately induce at test time, potentially introducing new attack surfaces rather than mitigating underlying risks.

With respect to novelty, reviewers expressed reservations about whether the contribution goes beyond existing or concurrent approaches. In the rebuttal, the authors explicitly acknowledge that the core methodology is the same as a concurrently submitted prior work, with differences primarily in experimental scope and analysis. Reviewers felt that this similarity, together with the framing of the paper as introducing a new technique, raises concerns that the novelty and methodological contribution may be overstated.

Taken together, reviewers found the empirical observations interesting but expressed reservations about the maturity of the framing, the strength of the novelty claims, and the extent to which the results support the broader conceptual and explanatory conclusions.

**Reviewer Concerns:**

Concerns Addressed by the Rebuttal

The rebuttal usefully clarified several points raised by reviewers. In particular, the authors acknowledged important limitations of the proposed approach, including that inoculation prompting does not confer full immunity to undesired behaviors and primarily affects controllability through contextual framing. The rebuttal also clarified aspects of the experimental setup, including how inoculation was applied across different datasets, and provided additional explanation regarding the interpretation of results in Appendix E.2. These clarifications helped address concerns about experimental ambiguity and improved the overall transparency of the work.

The rebuttal further acknowledged the relationship between the proposed approach and concurrent prior work, explicitly stating that the core methodology is shared, with differences lying mainly in experimental scope and analysis. This clarification helps situate the paper more accurately within the existing literature.

Concerns That Remain Outstanding

Despite these clarifications, several concerns remain. Most notably, the rebuttal does not fully resolve whether the reported effects can be largely explained by distributional or contextual differences between training and testing, rather than by a new learning principle. Although the authors describe this distributional shift as a deliberate feature of the approach, the rebuttal does not fully establish that simpler alternative explanations can be excluded.

Additionally, concerns about the safety implications of this approach persist. Although the authors acknowledge that inoculation may make certain behaviors more triggerable via prompts, the rebuttal primarily argues that this risk can be mitigated through filtering or input classifiers, rather than demonstrating that the issue is intrinsically addressed by the method itself.

Questions regarding novelty and positioning also remain partially unresolved. While the authors acknowledge methodological similarity to concurrent work, the paper continues to frame the approach as a distinct technique, and reviewers remain concerned that the novelty and contribution may be overstated relative to existing conditioning-based approaches.

**Reviewer Scores:**

Reviewer 3TsA (score 4) initially expressed significant concerns regarding the safety implications of inoculation prompting and the availability of simpler alternatives. In their post-rebuttal comment, the reviewer explicitly stated that they found the response helpful and updated their score accordingly, while still expressing some residual uncertainty about triggerability and stronger baselines. With full participation in the discussion, this reviewer would likely slightly increase their overall score.

Reviewer yepJ (score 4)  raised the most substantive conceptual concerns, particularly regarding distributional explanations, terminology, interpretation of experimental results, and novelty-related issues. While the rebuttal provided clarifications and acknowledged several limitations, it did not fundamentally resolve these core concerns. With full participation, I expect that this reviewer would likely maintain or slightly increase their original score, as the main conceptual objections remain largely intact.

Reviewer Tgu1 (score 6)  was positive and viewed the work as insightful. Their weaknesses focused on scope, interpretation, and presentation rather than core validity. Since the rebuttal did not introduce new concerns, I expect that their score would remain unchanged with full participation.

Reviewer m7rf  (score 6) was generally positive in the initial review, but raised several concerns regarding novelty, prompt-triggered vulnerabilities, and the interpretability of the toy experiments (e.g., capitalization versus response statistics).
In the rebuttal, the authors explicitly acknowledge that inoculated models can still be instructed to exhibit misaligned behavior at test time via injected system prompts, which confirms rather than resolves the reviewer’s concern about prompt-triggered vulnerabilities. The proposed mitigations (e.g., input filtering or combining with instruction hierarchy) are deployment-level or orthogonal techniques, rather than intrinsic solutions provided by inoculation prompting itself.
Additionally, the rebuttal does not directly address the reviewer’s diagnostic question regarding mixed capitalization in the training responses, which was intended to clarify what aspects of the signal the model is actually learning in the toy setting.

Given this, while the rebuttal improves transparency and positioning, it does not substantially alleviate the core method-level concerns raised by this reviewer. With full participation in the discussion, I expect that this reviewer would likely maintain a similar score, rather than increasing it.

---

### Decision · Program_Chairs · 2026-01-26

Accept (Poster)